

# Measurement Report: Water diffusion in single suspended phase-separated aerosols

Yu-Kai Tong[1], Zhijun Wu[2], Min Hu[2], and Anpei Ye[1]

[1]Key Laboratory for the Physics and Chemistry of Nanodevices, School of Electronics, Peking University, Beijing 100871, China
[2]State Key Joint Laboratory of Environmental Simulation and Pollution Control, College of Environmental Sciences and Engineering, Peking University, Beijing 100871, China

**Correspondence:** Anpei Ye (yap@pku.edu.cn)

**Abstract.** Water diffusion is a typical thermodynamic process in ambient aerosols which plays pivotal roles in their physico-chemical properties, atmospheric lifetime, and influences on the climate and human health. A fair amount of aerosols become phase-separated after experiencing atmospheric aging processes such as efflorescence, amorphization, and liquid-liquid phase separation. However, detecting the hygroscopicity of heterogeneous aerosols is quite intractable. Here, for the first time, we
directly characterize the water diffusion in single suspended phase-separated aerosols via a self-constructed laser tweezers Raman spectroscopy (LTRS) system. The $H_2O/D_2O$ isotope exchange is harnessed to trace the water diffusion in single laser-levitated homogenous/heterogeneous microdroplets. The time-resolved cavity-enhanced Raman spectra of the microdroplets is used to detect the diffusion process in real time. Two archetypes of phase-separated aerosols, i.e., partially engulfed and core-shell, are studied. Moreover, we quantify the dynamic water diffusion process by experimentally measuring the diffusion
coefficients. The results show that compared with the homogenous aerosols, water diffusion limitations exist in the phase-separated aerosols. The incomplete diffusion may stem from both the hydrophobicity of the organics and the formation of certain molecule clusters. This work provides possible implications on the evolutions, especially the gas-particle partition, of the actual phase-separated atmospheric aerosols.

## 1 Introduction

Gas-particle partitioning is one of the most significant atmospheric processes of aerosols which plays crucial roles in their impacts on air quality and atmospheric environment. As water is often the most mobile component in troposphere aerosols, a clear picture of water diffusion within aerosols is essential. Under various meteorological conditions, the size and refractive index of aerosols change via hydration and dehydration, which then influence the optical properties and ice nucleating ability of aerosols and the atmospheric energy distribution (Hallquist et al., 2009; Mellouki et al., 2015; Titos et al., 2016) . Besides,
water diffusion dictates the moisture content in aerosols and then impacts their component concentrations and phase states. Some previous works have shown that a substantial fraction of secondary organic aerosols (SOAs) have glassy or gel states which present slow heterogeneous reaction rates and nonequilibrium gas-particle partition(Bones et al., 2012; Fowler et al.,



2020; Shiraiwa and Pöschl, 2021) . It then may lead to significant kinetic constraints on aerosol processing, heterogeneous chemistry and component lifetimes (Renbaum-Wolff et al., 2013a; Shiraiwa et al., 2011; Vaden et al., 2011) .

Numerous techniques have been developed to study the hygroscopicity of aerosols, including electrodynamic balance (EDB), humidified tandem differential mobility analyzer (HTDMA), micro-Fourier transform infrared (FTIR) spectroscopy, atomic force microscopy, X-ray elemental microanalysis and attenuated total reflection FTIR spectroscopy (Kreidenweis and Asa-Awuku, 2014; Tang et al., 2019; Kuang et al., 2020) . In these techniques, four main methods are used to detect the water diffusion process. (i) The differential step isothermal method developed by Aristov et al. (Cai et al., 2015; Lv et al., 2020;
Tong et al., 2022a, b) circumvents the non-linear boundary value problem in analyzing water diffusion process and can readily retrieve the water diffusion coefficient by fitting the response of a single droplet to a changing relative humidity (RH) during sorption/desorption experiments. However, it can only be used to simulate the hygroscopic process of high viscosity droplets where water diffusion is quite slow and cannot apply to constant RH conditions. (ii) The Stokes-Einstein (S-E) equation relates the water diffusion coefficient to the particle viscosity. Many experimental and theoretic evaluation methods have been
developed to measure the viscosity of aerosol particles both in laboratory and in field (Sastri and Rao, 1992; Cao et al., 1993; Rothfuss and Petters, 2017; Booth et al., 2014; Maclean et al., 2021; Smith et al., 2021; Fitzgerald et al., 2016; Renbaum-Wolff et al., 2013b; Bishop et al., 2004) . However, application of the S-E equation in tandem with viscosity measurements may also miscalculate the diffusion coefficient because the S-E equation have been shown to break drown at high viscosities (Power et al., 2013; Molinero and Goddard, 2005) . (iii) Another method leverages the response of aerosols to the oscillating
RH to retrieve the diffusion coefficient. The exploited RH is regulated to oscillate in pulse form (Leng et al., 2015; Shi et al., 2017) or sinusoidal form (Preston et al., 2017) . For a sinusoidal RH oscillation, the amplitude and frequency of the aerosol size fluctuation are dictated by the RH frequency and the diffusion coefficient of water molecules. Nonetheless, this method demands a highly-sensitive and precise RH control system, which increase the complexity of the experiments. (iv) The isotopic tracer method can directly unveil the water diffusion process of aerosol droplets, where the deuterium oxide ($D_2O$) molecules
are leveraged to trace the diffusion of water within hydrogen oxide ($H_2O$) microdroplets (Price et al., 2014; Davies and Wilson, 2016; Moridnejad and Preston, 2016; Nadler et al., 2019) . One prominent advantage of this method is that it is available to study the water diffusion process at constant RH conditions, where the chief driving force of diffusion is the concentration gradient rather than RH changes, while the aforementioned methods can only study the hygroscopic response of aerosols to RH changes.

Previous works mainly focused on the hydration/dehydration of homogenous aerosols. However, a plethora of studies have shown that phase separation is prevalent in ambient aerosols (You et al., 2014; Freedman, 2017, 2020; Pöhlker et al., 2012; You et al., 2012; Lee et al., 2020) . Modeling works show that ignoring phase separation by forcing a single non-ideal phase, can lead to vastly incorrect gas-particle partitioning predictions (Pye et al., 2017; Zuend and Seinfeld, 2012) . Indeed, it is now widely recognized that the existence of heterogeneous states (e.g., phase-separated and amorphous states) could have significant
consequences for the composition of the condensed aerosol phase. For example, the isoprene-derived SOAs are typical phase-separated aerosols which are formed by heterogeneous reactive uptake of epoxydiols onto sulfate aerosol particles. Some works reported that the growth of the SOA coatings may impede the reactive uptake of epoxydiols, rendering a self-limiting effect





in isoprene-derived SOAs formation (Zhang et al., 2018, 2019; Riva et al., 2019) . The similar diffusion limitation is also
observed in the uptake of $\alpha$-pinene oxide into acidic aerosols (Drozd et al., 2013) and in the ozonolysis of polycyclic aromatic
hydrocarbons within SOAs (Zhou et al., 2019) . For water diffusion, Davies et al. (2013) found that organic coatings (long-
chain alcohols) may reduce the evaporation of the aerosol liquid water and enhance the condensation of water on the droplets.
Other works found that the water condensation is hampered by organic shells and the hygroscopic growth of phase-separated
aerosols are dependent on the thickness of shells (Ruehl and Wilson, 2014; Li et al., 2021; Mikhailov et al., 2021) . However,
some other works reported that phase separation has no profound effect on the water diffusion under normal ambient conditions
(Chan et al., 2006; Zawadowicz et al., 2015; Lienhard et al., 2015) .

Notwithstanding, nearly all previous works used the substrate-deposited samples to study mass transfer in phase-separated
aerosols. Contrastingly, contactless single particle techniques are appealing, because the impacts of surface perturbations on
component concentrations and aerosol morphology can be excluded (Zhou et al., 2014) . In addition, single particle mea-
surements are preferred over ensemble-averaged experiments, because composition and local chemical environments vary
from particle to particle. In this work, we utilize isotope tracing to characterize the water diffusion process in single sus-
pended phase-separated aerosols at constant RH and room temperature via a self-constructed laser tweezers Raman spec-
troscopy (LTRS) system. The time-resolved cavity-enhanced Raman spectra of the microdroplets is recorded to both detect
the phase state and reveal the diffusion of water. Three types of aerosols are herein studied, including homogenous aerosols
($D_2O$+citric acid (CA)), partially engulfed aerosols ($H_2O$+ammonium sulfate (AS)+oleic acid (OA)), and core-shell aerosols
($H_2O$+AS+diethyl-L-tartrate (DLT) and $H_2O$+AS+1,2,6-hexanetriol (HEX)). Moreover, the influence of acid on water diffu-
sion in aerosols is also discussed.

## 2   Experimental and Methods

### 2.1   Laser tweezers Raman spectroscopy system

A schematic of the LTRS system is shown in Fig. 1. A laser beam with a wavelength of 532 nm (Excelsior-532-200, Spectra
Physics) is used as both trapping and Raman exciting light. The backscattering Raman light is conducted into a spectrograph
(SpectaPro 2300i, Acton) equipped with a liquid nitrogen cooled CCD (Spec-10, Princeton Instruments) working at a temper-
ature of -120 °C.

Bulk solutions with desired chemical compositions are used to generate the aerosol droplets by a medical nebulizer (Mint
PN100). In a tailored aerosol trapping chamber (see Supplementary Fig. S1), individual droplets (4~10 $\mu$m) from an incoming
droplet train will be trapped and levitated by the laser tweezers. More details of the LTRS system can be seen in our previous
works (Tong et al., 2022a, b, c) . For $D_2O$+solute aerosols, a $D_2O$ bubbler is first used to provide moisture in the trapping
chamber; after the droplet equilibrates to the surrounding water vapor, the flow path is turned to a $H_2O$ bubbler by 3-way
valves to observe the substitution of $H_2O$ for $D_2O$ within the droplet. For $H_2O$+solute aerosols, the moisture is first provided
by $H_2O$ bubbler and then by $D_2O$ bubbler and the substitution process of $D_2O$ for $H_2O$ is studied.





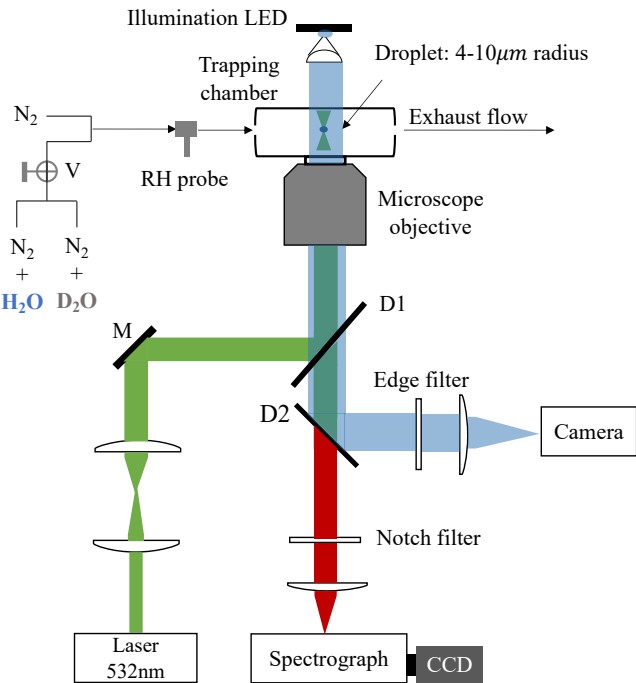

**Figure 1.** Schematic of the laser tweezers Raman spectroscopy system. A 532 nm laser beam is used to both trap the aerosol droplet and excite the Raman signal. The droplet is imaging using a 470 nm illumination LED and a high frame rate camera. The Raman spectra of the trapped droplets are recorded using a spectrograph/CCD. The RH in the aerosol trapping chamber is regulated by a flow of mixing $N_2/N_2 + H_2O$ or $N_2/N_2 + D_2O$. V is 3-way valves. M is a mirror. D1 and D2 are dichroic mirrors.

## 2.2 Detection of phase separation

The phase separation in substrate-deposited aerosols can be directly observed through bright-field imaging, TEM (transmission electron microscopy), or TXM (transmission X-ray microscopy) (Pöhlker et al., 2012; You et al., 2012; Lee et al., 2020; Ma et al., 2021) . However, for levitated droplets, the defocus of the trapped droplets blurs the direct imaging. Instead, the time-resolved Raman spectra of the trapped droplets can be used to efficiently detect phase separation (Tong et al., 2022c; Gorkowski et al., 2016, 2018, 2020; Sullivan et al., 2020) .

The trapped droplet works as an enhancing cavity and will overlap stimulated sharp peaks at wavelengths commensurate with whispering gallery modes (WGMs) on the spontaneous Raman spectra. The Raman spectra of aerosols with three different morphology archetypes are shown in Supplementary Fig. S2. The spectra containing high-quality WGMs indicates the isotropy within the particle and yields a homogenous morphology. The spectra containing weak but noticeable WGMs indicates the symmetry of the particle remains and yields a core-shell morphology. Meanwhile, the spectra without any WGMs indicates the destruction of both isotropy and symmetry in the particle and yields a partly engulfed morphology. Alternatively, Stewart et al.





(2015) put forward another two signatures to detect phase separation in aerosol. One is that if the droplet radius and refractive index calculated by Mie scattering model present an abrupt change, which is not realistic, it means that phase separation has occurred. The other is that if we fit the Raman spectra with the Mie scattering model for a homogenous droplet and the fitting

errors between the measured and simulated WGM peaks increase by orders of magnitude, the droplet can be determined as inhomogeneous. Herein, we deploy the signatures of WGMs and fitting errors to detect the phase separation. The homogenous Mie scattering fitting model used in this work was developed by Preston and Reid (2015).

## 3   Results

Here, we first detect the water diffusion in homogenous droplets to validate the performance of the isotope trace method.

Then, the water diffusion in $H_2O+AS+OA$, $H_2O+AS+DLT$, and $H_2O+AS+HEX$ droplets is studied. The diffusion differences in these aerosols with different morphologies are discussed. Moreover, by adding sulfuric acid to $H_2O+AS+DLT$ droplets, we also detect the influence of proton on water diffusion in aerosols is also discussed.

### 3.1   Raman spectra snapshots during water diffusion

Although $H_2O$ and $D_2O$ have nearly identical physical properties, O-D and O-H have different energy levels, which are

therefore characterized with disparate Raman shifts (see the spectra of bulk $H_2O$ and $D_2O$ solutions in Supplementary Fig. S3). Thus, the rise and fall of O-D/O-H peaks in Raman spectra can be used to trace water diffusion.

Fig. 2 presents the representative stills of the Raman spectra of $H_2O+AS+DLT$ droplet at different water diffusion progressions. The brand range of 640~660 nm corresponds to the bending and stretching modes of O-H of water, the band in range of 605~625 nm corresponds to the modes of O-D , and the range of 627~635 nm corresponds to the bending mode of C-H in

organics (DLT here). It can be seen that at the early stage (t = 1 min) of water diffusion, the $\nu$(O-H) is vastly predominant and the $\nu$(O-D) is quite trivial. As water diffusion progressing (t = 40 min), the intensity of $\nu$(O-D) mode rises while $\nu$(O-H) mode falls. It indicates that with the surrounding moisture vapor being switched from $H_2O$ to $D_2O$, the $H_2O$ molecules within the droplet are being replaced by $D_2O$ molecules, albeit at a constant RH condition. For t = 100 min, $\nu$(O-D) becomes predominant compared with $\nu$(O-H), indicating that the droplet has changed from a $H_2O$ droplet to a $D_2O$-dominating droplet. Compared

with the Supplementary Fig. S3, it can be seen that both $\nu$(O-H) and $\nu$(O-D) modes in suspended aerosols are weaker than that in corresponding bulk solutions, which means the total water content in aerosols is far lower than that in their mother solutions. It underscores the advantage of this contactless single technique that without the surface perturbations, the component concentration in the aerosol can exceed its solubility limit.

At a constant RH condition, the total amount of water ($D_2O$ plus $H_2O$) in the aerosol can be supposed to remain constant.

This prerequisite has been confirmed in Fig. 6. Thus, the time-resolved fractional concentration of $D_2O$ (denoted by $\phi_{OD}$) can be calculated from the $\nu$(O-H) and $\nu$(O-D) modes at each spectral time:

$$\phi_{OD} = \frac{A_{OD}}{A_{OD} + \frac{1}{\sqrt{2}}A_{OH}}, \tag{1}$$



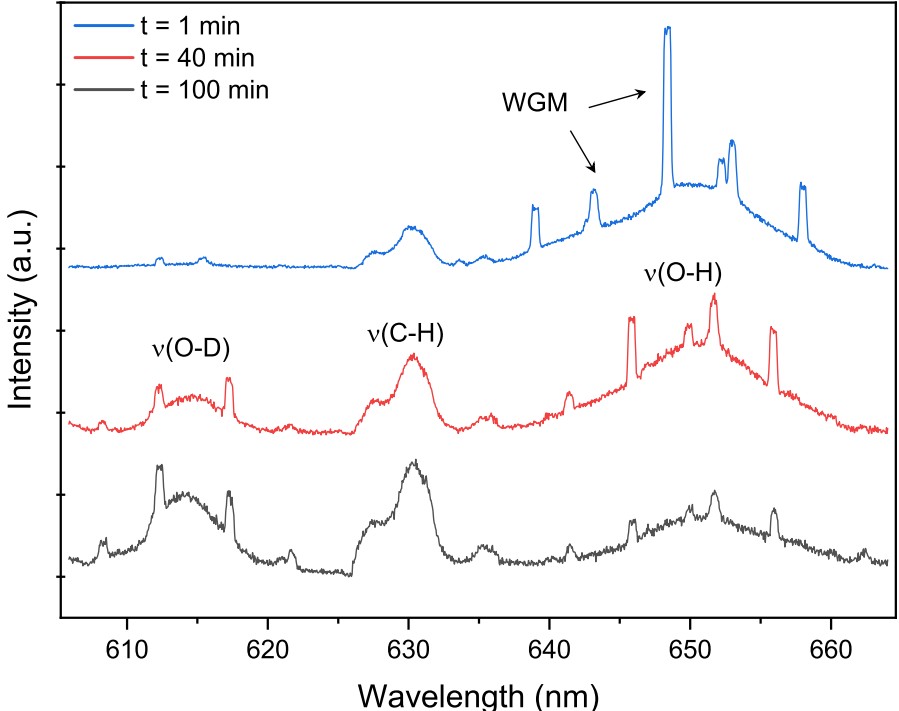

**Figure 2.** Raman spectra snapshots of $H_2O$+AS+DLT aerosol at different time during the water diffusion process. The RH in the trapping chamber is constant as 60%. Blue, red and black curves indicate the Raman spectrum extracted from Fig. 5(A) at 1, 40 and 100 min, which correspond to the initial, middle and end stage of water diffusion process, respectively. The WGMs and Raman feature bands are pointed out. t = 0 means the onset of switching $H_2O$ moisture vapor to $D_2O$ vapor.

where $A_{OD}$ and $A_{OH}$ are the integrated intensities of $\nu$(O-D) and $\nu$(O-H) modes respectively. The factor of $1/\sqrt{2}$ before $A_{OH}$ is to compensate the difference in reduced mass between hydrogen and deuterium (Price et al., 2014; Nadler et al., 2019) .

Therefore, the temporal variations of $\phi_{OD}$ retrieved from the aerosol Raman spectra can be used to quantify the water diffusion process. A caveat is that Fig. S4 shows the calculated $\phi_{OD}$ after effacing WGMs in the spectra, indicating that the contribution of WGMs to the peak areas is inconsequential. Thus, the presented $\phi_{OD}$ hereafter is calculated with ignoring the WGMs influences.

### 3.2 Water diffusion in homogenous aerosols

The water diffusion of single $D_2O$+CA aerosol exposed to $H_2O$ moisture vapor is shown in Fig. 3. The droplet was first trapped and equilibrated in $D_2O$ ambiance. At t = 0, the gas manifold valves are rotated to switch from $D_2O$ to $H_2O$. In Fig. 3(A), It can be seen that, over the time, the intensity of $\nu$(O-D) deteriorates rapidly and that of $\nu$(O-H) increases. Meanwhile, the intensity of $\nu$(C-H) keeps stable which indicates that the component concentration in the aerosol is constant throughout the experiment. The existing WGMs in each spectrum means that the droplet is spherically symmetric.





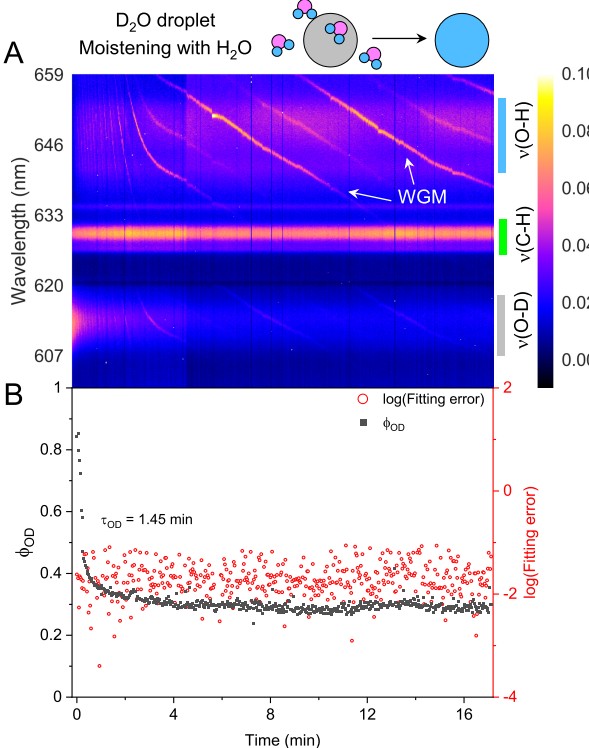

**Figure 3.** Water diffusion in single homogenous $D_2O$+CA aerosol at RH = 60%. (A) The time-resolved cavity-enhanced Raman spectra of the droplet. The abscissa is time and the ordinate indicates the wavelength. The spectral intensity at each wavelength and time is illustrated with color. The WGMs are pointed out with white arrows. The top diagram depicts the water diffusion process, where light grey represents $D_2O$ phase and blue represents $H_2O$ phase. The modes of $\nu$(O-H), $\nu$(C-H), and $\nu$(O-D) are pointed out with different color bars on the right. (B) Black, the temporal variation of fractional concentration of $D_2O$ within the droplet; red, the fitting errors of the WGMs based on the homogenous Mie scattering model. $\tau_{OD}$ is the $e$-folding time of the $\phi_{OD}$ curve. t = 0 means the onset of switch $D_2O$ moisture vapor to $H_2O$ vapor. The experiment was conducted at room temperature.

The temporal variation of $\phi_{OD}$ in Fig. 3(B) vividly shows the substitution of $H_2O$ for $D_2O$. The calculated $e$-folding time of the $\phi_{OD}$ curve (formally $\tau_{OD}$) is 1.45 min, indicating that the homogenous aerosol can promptly response to the variation of surrounding atmosphere. Fig. 3(B) shows that the fitting errors of the measured WGMs calculated by the homogenous Mie scattering model are on the order of $10^{-2}$, which are pretty small (compared with Fig. 7). It means that the droplet is well-mixed and isotropic. The CA is a water soluble organic compound, thus the suspended $D_2O$+CA aerosol is homogenous and has a spheric shape, which is validated by both the WGMs in spectra and the fitting errors.



### 3.3 Water diffusion in partly engulfed aerosols

The oleic acid is a preferential proxy of water insoluble organics in ambient aerosols. Here, we nebulized a mixed solution containing of AS, OA and $H_2O$ and trapped an aerosol droplet to observe the water diffusion in such phase-separated droplet. The droplet was trapped and equilibrated in $H_2O$ ambiance. Then the vapor was switched from $H_2O$ to $D_2O$.

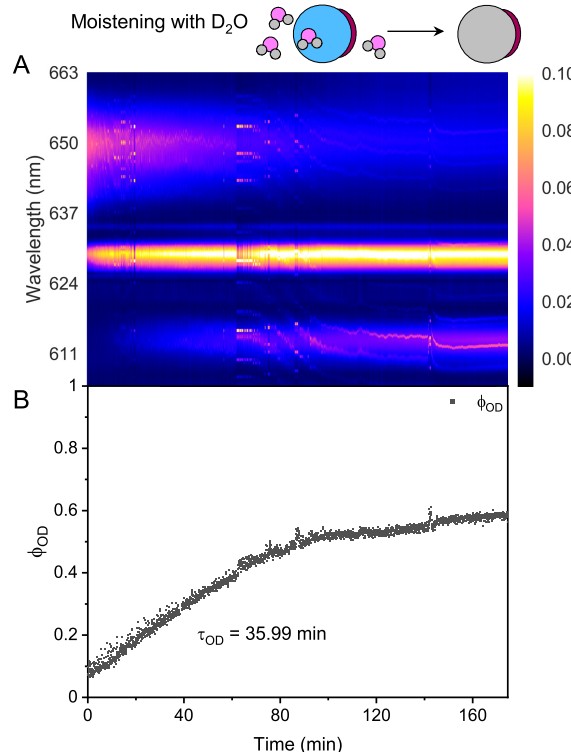

**Figure 4.** Water diffusion in single phase-separated $H_2O$+AS+OA aerosol at RH = 60%. (A) The time-resolved Raman spectra of the droplet. The top diagram depicts the water diffusion process in the partly engulfed aerosol, where light grey represents the $D_2O$ phase, blue represents the $H_2O$ phase, and dark red represents the hydrophobic organic phase. (B) Black, the temporal variation of fractional concentration of $D_2O$ within the droplet. t = 0 means the onset of switch $H_2O$ moisture vapor to $D_2O$ vapor. The experiment was conducted at room temperature.

Fig. 4 shows the water diffusion in single $H_2O$+AS+OA aerosol. Fig. 4(A) shows no evident WGMs in the aerosol spectral, indicating the destruction of both isotropy and symmetry in the particle. Thus, the droplet should have a partly engulfed morphology after reaching a thermodynamic equilibrium with the surrounding moisture, where a hydrophobic cap of OA encases an aqueous phase. The spectral variation at t = ∼60 min may stem from the drift of the hydrophobic cap because the hydrophobic phase is not always at the bottom of the droplet (Ishizaka et al., 2021) . Besides, the volume ratio of aqueous phase

and hydrophobic phase in the trapped droplet cannot be preset because of the stochastic mixing of OA emulsions and water during nebulizing. If an approximately spherical cavity occurs for the aqueous volume, the WGM fingerprint of the droplet





may exhibit low quality and complexity. Moreover, the band of C-H here is stronger than that in Fig. 3(A) which may result from that the OA molecule has more C-H bonds than CA.

The $\phi_{OD}$ shown in Fig. 4(B) changes dramatically more slowly than that in Fig. 3(B). The calculated $\tau_{OD}$ of $H_2O$+AS+OA
aerosol is $\sim 35.99$ min which is 25 times of $D_2O$+CA. It means that an inhibition of gas-particle partitioning occurs in such phase-separated droplet. The OA phase in the droplet has a considerably strong hydrophobicity which may impede the moisture diffusing through the organic cap. The effective interface between the aqueous phase and the air reduces because of the phase separation, leading to a slower water diffusion compared with the homogenous aerosol.

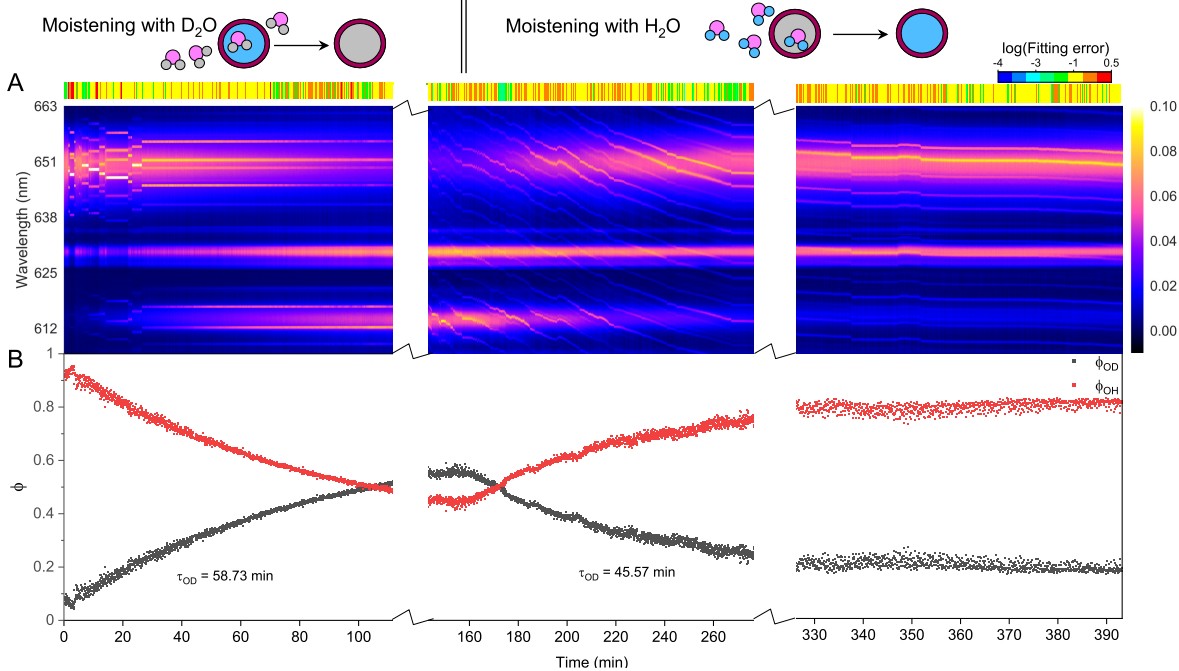

**Figure 5.** Water diffusion in single phase-separated $H_2O$+AS+DLT aerosol at RH = 60%. (A) The time-resolved cavity-enhanced Raman spectra of the droplet. The top color bars indicates the log(fitting errors) of the WGMs based on the homogenous Mie scattering model. The running of the spectrograph needs a break to avoid overloading the shutter, which causes the hiatuses in the spectra. (B) Black, the temporal variation of fractional concentration of $D_2O$ within the droplet; red, the temporal variation of fractional concentration of $H_2O$ within the droplet, $\phi_{OH} = 1 - \phi_{OD}$. The droplet was first trapped and equilibrated in $H_2O$ ambiance. t = 0 means the onset of switch $H_2O$ moisture vapor to $D_2O$ vapor. At t = 160 min, the moisture vapor was switched back from $D_2O$ to $H_2O$. The experiment was conducted at room temperature.

## 3.4 Water diffusion in core-shell aerosols

The core-shell morphology is another prevail phase-separated morphology of ambient aerosols. Here, we generated aerosol droplets from the mother solutions containing $H_2O$+AS+DLT and $H_2O$+AS+HEX and induce phase separation in them by




presetting the surrounding RH below their separation relative humidity. The droplets were trapped and equilibrated in $H_2O$ ambiance before switching the moisture vapor from $H_2O$ to $D_2O$.

Fig. 5 panoramically presents a panorama of the diffusion of $D_2O$ and $H_2O$ in single $H_2O$+AS+DLT droplet during a 7-hour
observation at RH = 60%. As shown in Fig. 5(A), the log(fitting errors) throughout the observation is roughly higher than -1 which is one order higher than the homogenous aerosol errors, indicating that the droplet is not homogenous. In addition, the WGMs persist in the whole observation, thus the droplet should be core-shell.

To provide detailed insights into the phase-separated structure, we use a core-shell Mie model developed by Vennes and Preston (2019) to calculate the core and shell radius of the droplet. In Fig. 6, it can be seen that for the spectra shown in
Fig. 5(A) (t = 0~110 min), the calculated particle radius is around 5 $\mu$m and the fluctuation is quite trivial. Meanwhile, the calculated radius ratio (i.e., the ratio of the core radius to the whole particle radius) is around 0.8, which yields a core radius of 4 $\mu$m and a shell thickness of 1 $\mu$m. The results verify that at constant RH conditions, the switch of moisture vapor from $H_2O$ to $D_2O$ does not change the size of the aerosol. Besides, it also validates that at RH = 60%, the liquid-liquid phase separation occurs in the $H_2O$+AS+DLT aerosol and the separated morphology is core-shell.

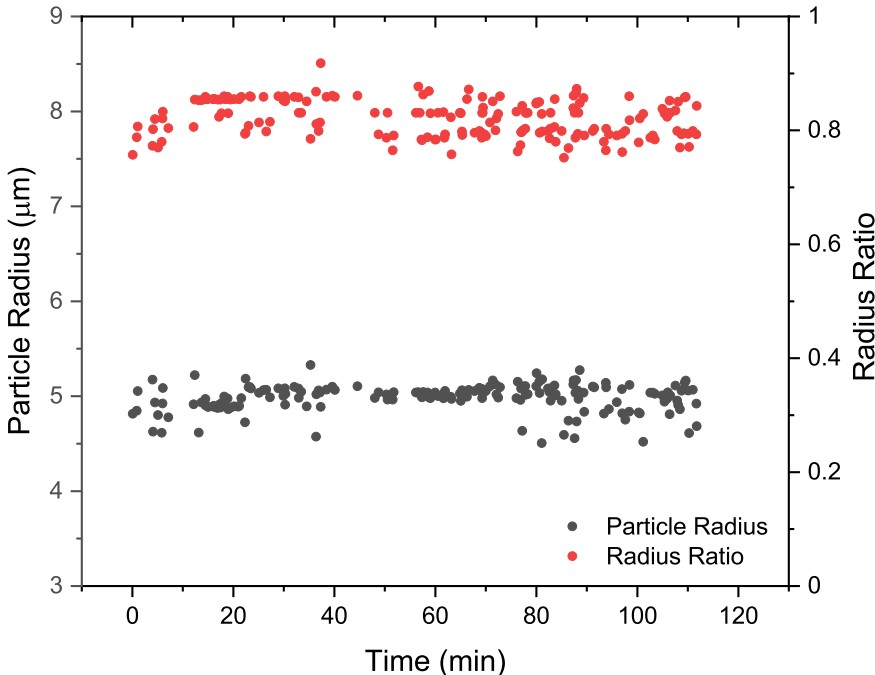

**Figure 6.** Radius of the phase-separated $H_2O$+AS+DLT aerosol at RH = 60%. Black, the aerosol radius; red, the ratio of the core radius to the whole particle radius. The results are obtained by fitting the spectra in Fig. 5(A) (t = 0~110 min) with a core-shell model developed by Vennes and Preston (Vennes and Preston, 2019) .

For t = 0~160 min (denoted by Stage I), the droplet was moistened by $D_2O$ vapor and $D_2O$ molecules started to diffuse into the droplet. The intensity of O-H band in Fig. 5(A) decreases while the O-D band increases; this variation is crystal



clear to be seen in Fig. 5(B) where $\phi_{OD}$ grows and $\phi_{OH}$ falls over the time. However, in this stage of D$_2$O diffusing into the droplet, the $\phi_{OD}$ plateaus to a constant value of 0.55 after more than 150-minute diffusion, indicating that the H$_2$O molecules in the initial droplet cannot be replaced completely by the surrounding gas-phase D$_2$O molecules. The intensity of O-H band cannot diminish to zero in the exchange process. Similar results can also be seen in the partly engulfed aerosol and even homogenous aerosol, where the initial values of $\phi_{OD}$ are 0.6 and 0.3 respectively. Previous works have reported such kinetic limitations of diffusion in ultra-viscous or amorphous state aerosols where significant radial gradients in pH (Wei et al., 2018), solute concentrations (Zobrist et al., 2008), and reactant uptake (Virtanen et al., 2010; Davies and Wilson, 2015; Gaston and Thornton, 2016) solidly exist in ambient aerosols. A possible explanation is that certain molecule clusters composed of hydroxyls, electrolytes, and organics forms in the aerosols because of supersaturation, which protects a handful of H$_2$O molecules in the aerosols from being replaced by the D$_2$O molecules. Moreover, with the progressing of water diffusion, the diffusion-driving forces attenuate because of the reducing deviation of concentrations between gas and particle phase, which may both decrease the success of surface accommodation of gas molecules and make the solvation through the particle bulk more difficult, rendering an impossible complete molecules substitution.

For t > 160 min (denoted by Stage II), the moisture vapor was switched back from D$_2$O to H$_2$O. In Fig. 5(A), it can be seen that the Raman band of O-H rebounds and that of O-D declines over the time. After molecules diffusing for 4 hours, $\phi_{OD}$ does not diminish to zero and $\phi_{OH}$ does not return to 1, yielding a similar incomplete substitution. Noteworthily, in Fig. 5(B), throughout Stage I and II, the maximum of $\phi_{OH}$ is 0.8 which is higher than that of $\phi_{OD}$. It indicates that, at the later stage of diffusion, D$_2$O is harder to partition into the particle phase than H$_2$O. Considering the virtually identical chemistry of these two molecules, one may think that the difference of molecular mass gives rise to the different final diffusion extent. In another perspective, during the process of aerosol trapping, the generated aerosol train may condense some droplets on the walls of the chamber and tubes, the H$_2$O molecules in which may interfere the subsequent water diffusion.

As shown in Fig. 5(B), the $\tau_{OD}$ of Stage I and II are 58.7 min and 45.6 min respectively, which are both higher than that of partly engulfed aerosol and homogenous aerosol. The averaging $\tau_{OD}$ is 52.2 min which is 1.5 times of the partly engulfed aerosol and 36 times of the homogenous aerosol, implying a more profound diffusion inhibition in core-shell aerosols. With the organic shell totally encasing the aqueous core, the moisture molecules have to penetrate through the shell during diffusion, which vastly retards the molecules exchange.

We then observed the water diffusion in single H$_2$O+AS+HEX aerosol. Fig. 7 shows the recorded Raman spectra and fractional concentration variations. In the initial period, the capricious but noticeable WGM fingerprint may stem from the surface fluctuation of the droplet due to capillarity (Endo et al., 2018; Chung et al., 2017; Pigot and Hibara, 2012) . At t = 0, the H$_2$O droplet started to be moistened by D$_2$O vapor. The results of log(fitting errors) and spectral WGMs indicates the droplet was phase-separated with a core-shell morphology throughout the observation. With the droplet being exposed to the D$_2$O vapor, the Raman O-H band diminishes and the O-D band rises. However, as shown in Fig. 7(B), the calculated $\tau_{OD}$ is 88.7 min, implying a severer diffusion inhibition even compared with the H$_2$O+AS+DLT aerosol. It may be attributed to that HEX molecules are smaller than DLT molecules so that the shell of HEX is more compact than that of DLT. Thus, the pores





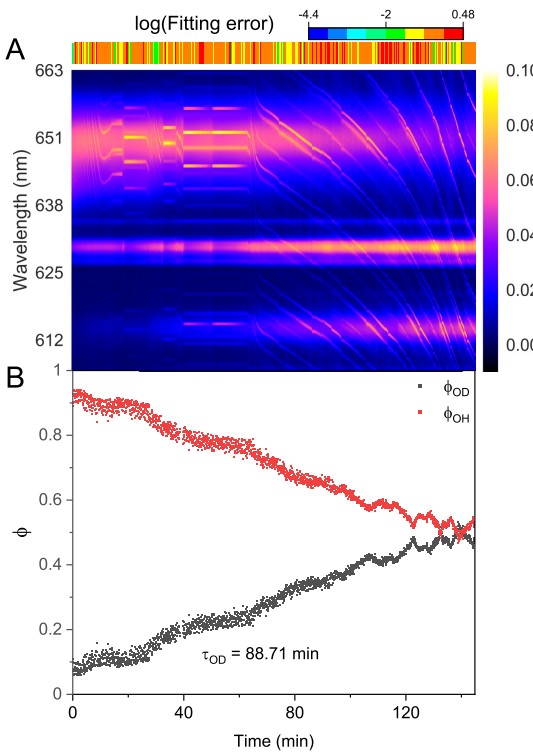

**Figure 7.** Water diffusion in single phase-separated $H_2O$+AS+HEX aerosol at RH = 60%. (A) The time-resolved cavity-enhanced Raman spectra of the droplet. The top color bars indicates the log(fitting errors) of the WGMs based on the homogenous Mie scattering model. (B) The temporal variations of fractional concentration of water molecules within the droplet. The droplet was first trapped and equilibrated in $H_2O$ ambiance. t = 0 means the onset of switch $H_2O$ moisture vapor to $D_2O$ vapor. The experiment was conducted at room temperature.

and channels formed in HEX shell are smaller and more difficult for water molecules to pass through. Besides, the HEX has a stronger hydrophobicity than DLT, which makes the diffusion through HEX shell more difficult.

The surplus protons are considered to have appreciable impacts on the phase separation in ambient aerosols (Tong et al., 2022c; Dallemagne et al., 2016; Losey et al., 2016) . Here, we added sulfuric acid to $H_2O$+AS+DLT droplets and observed

the water diffusion process in the resultant acidified aerosols. The pH of the mother solution to generate aerosols was preset to 1.17. Fig. 8 shows the recorded Raman spectra and fractional concentration variations of single acidified $H_2O$+AS+DLT droplet. The results of log(fitting errors) and spectral WGMs indicates that the droplet was homogenous. After being moistened by $D_2O$ vapor at t = 0, the Raman O-H band fades and the O-D band grows up. The $\tau_{OD}$ shown in Fig. 8(B) is 17.4 min which is less than the value of the two types of phase-separated aerosols. It shows that the surplus protons increase the rate of

water diffusion in $H_2O$+AS+DLT aerosols, which indicates that the added sulfuric acid may impede the occurrence of phase separation. A possible explanation may be that the excess protons from the added sulfuric acid may enhance the polarity of organic molecules and increase the miscibility between the organic component and water.





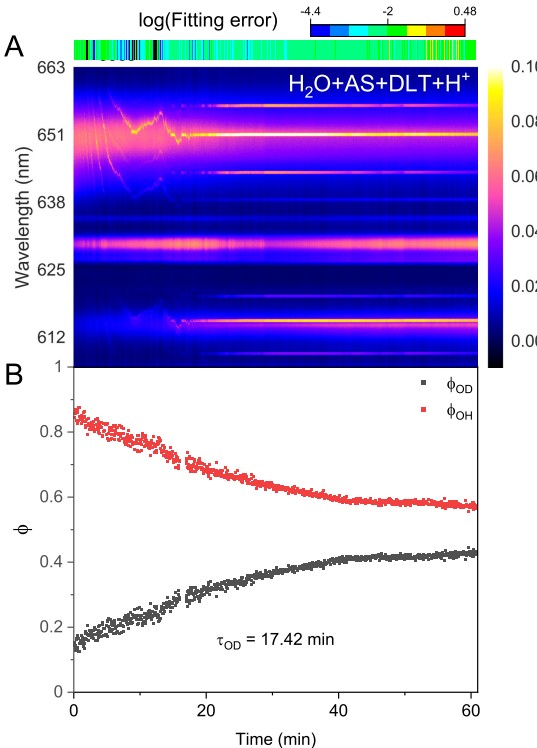

**Figure 8.** Water diffusion in single acidified $H_2O$+AS+DLT aerosol at RH = 60%. (A) The time-resolved cavity-enhanced Raman spectra of the droplet. The top color bars indicates the log(fitting errors) of the WGMs based on the homogenous Mie scattering model. (B) The temporal variations of fractional concentration of water molecules within the droplet. The droplet was first trapped and equilibrated in $H_2O$ ambiance. t = 0 means the onset of switch $H_2O$ moisture vapor to $D_2O$ vapor. The experiment was conducted at room temperature.

## 4  Discussion

The isotope exchange during the water diffusion process in single aerosols can be well elucidated by the solution to Fick's

second law for a sphere (Price et al., 2014; Moridnejad and Preston, 2016; Nadler et al., 2019) :

$$\phi_{OD} = 1 - \left(\frac{6}{\pi^2}\right) \sum_{n=1}^{\infty} \frac{1}{n^2} \exp\left(-\frac{n^2\pi^2 D_w t}{a^2}\right), \tag{2}$$

where $a$ is the particle radius and $D_w$ is the diffusion coefficient of water. A prerequisite of applying the Fickian diffusion model is that the particle achieves a homogenous mixture after sufficient equilibration time, which is however not available for the aerosols studied here. Thus, a modified Fickian diffusion model is used here to analyze the observed incomplete isotope

exchange, where a correction factor $\chi$ is introduced that reveals the diffusion limitation and means the diffusing extent (see Eq. 3).

$$\phi_{OD} = \chi\left[1 - \left(\frac{6}{\pi^2}\right) \sum_{n=1}^{\infty} \frac{1}{n^2} \exp\left(-\frac{n^2\pi^2 D_w t}{a^2}\right)\right]. \tag{3}$$



**Table 1.** Water diffusion coefficients of aerosols with various morphologies at RH = 60%.

| Aerosol | Morphology | $\tau_{\mathrm{OD}}$ (min) | $\chi$ | $D_{\mathrm{w}}$ ($\times 10^{-16} m^2 s^{-1}$) |
|---|---|---|---|---|
| $D_2O$+CA | homogenous | 1.45 | N/A | N/A |
| $H_2O$+AS+OA | partly engulfed | 35.99 | 0.77±0.013 | 13.18±0.63 |
| $H_2O$+AS+DLT (Stage I) | core-shell | 58.73 | 0.73±0.016 | 5.36±0.33 |
| $H_2O$+AS+HEX | core-shell | 88.71 | 0.65±0.029 | 2.25±0.15 |
| $H_2O$+AS+DLT+$H^+$ | homogenous | 17.42 | 0.49±0.004 | 39.96±0.11 |

The radii of the aerosols can be determined by bright-field imaging and the core-shell Mie model. Then, the $D_{\mathrm{w}}$ can be expediently derived by using the three-term expansion of the modified Fickian diffusion model to fit the temporal variations of

$\phi_{\mathrm{OD}}$ retrieved from the Raman spectra. An application of the diffusion model to the isotope exchange data is shown in Supplementary Fig. S5. The water diffusion coefficients of the aforementioned aerosols are summarized in Tab. 1. The comparison between the measured water diffusion coefficients in this work and literature works can be seen in Supplementary Tab. S1. For $H_2O$+CA droplet, according to the results of Davies and Wilson (2016) , the calculated $\tau_{\mathrm{mixing}}$ is ∼20 s which is in the same order magnitude as the observed $\tau_{\mathrm{OD}}$ (∼80 s) in this work. For the phase-separated droplets, the measured $D_{\mathrm{w}}$ is considerably

lower than the values of homogenous droplets studied in literature works, which indicates the occurrence of the water diffusion limitations.

The isotope exchange method has some experimental limitations. For example, the spectral acquisition costs time; after switching $H_2O$ to $D_2O$, it also takes time to fully replace the composition of the atmosphere in the trapping chamber. Due to these inevitable time limitations, the isotope exchange method does not adapt to quantifying the rapid diffusion circumstance,

leading to an upper limit of diffusion coefficient measuring of $\sim 10^{-13} m^2 s^{-1}$ (Davies and Wilson, 2016; Nadler et al., 2019). The water diffusion in $D_2O$+CA droplet is quite fast, hence the onset of water diffusion cannot be exactly determined. The observed $\tau_{\mathrm{OD}}$ maybe underestimate the rate of water diffusion. Thus, the $D_{\mathrm{w}}$ of the $D_2O$+CA aerosol is not calculated in Tab. 1. The $H_2O$+AS+OA aerosol is treated as an approximate sphere. The measured $D_{\mathrm{w}}$ decreases in the order of homogenous, partly engulfed, and core-shell aerosols, which is inline with the diffusion rate presented in Section 3.

The parameter of the first exponential term in Eq. 2 (i.e., $\pi^2 D_{\mathrm{w}}/a^2$) indicates the rate of diffusion for a homogenous aerosol, the reciprocal of which means the equilibrium mixing time (denoted by $\tau_{\mathrm{mixing}}$) of the volatile molecules within the homogenous aerosol. According to the modified Fickian diffusion model, the fitted $\tau_{\mathrm{mixing}}$ of $H_2O$+AS+DLT is 182 min (see the Supplementary Fig. S5). However, in Fig. 5(B), it can be seen that when t = 130 min, the $\phi_{\mathrm{OD}}$ levels off which means the diffusion of $D_2O$ has reached a balance. The observed equilibrium diffusion time is less than the calculated $\tau_{\mathrm{mixing}}$. It implies

that the water molecules within these aerosols do not diffuse to an isotropically stable state, thus the concentration gradients exist in the aerosols. It revalidates the deductions from the temporal variations of $\phi_{\mathrm{OD}}$. If considering the correction factor $\chi$, the $\chi\tau_{\mathrm{mixing}}$ of $H_2O$+AS+DLT is 132 min which agrees well with the experimental observation. It means the modified Fickian diffusion model works well to simulate the water diffusion presented here.



The $\chi$ of $H_2O$+AS+DLT is 0.7, which means that 70% of the total $H_2O$ molecules in the droplet are substituted by $D_2O$
molecules. The $\chi$ of $H_2O$+AS+HEX is 0.65 which indicates that the more condensed shell of HEX leads to a lower diffusion
extent of $D_2O$ than $H_2O$+AS+DLT. Contrastingly, the $\chi$ of $H_2O$+AS+OA aerosol is 0.77 which indicates that the partly
uncovered gas-particle interface allows for a higher diffusion extent of $D_2O$ than the core-shell aerosols. Of note, the $\chi$ of
acidified $H_2O$+AS+DLT aerosol is 0.5 which is lower than the partly engulfed and core-shell aerosols. A fair amount of
studies have reported the existence of hydrated proton clusters with diverse structures in acid solutions (Headrick et al., 2005;
Biswas et al., 2017; Knight and Voth, 2012; Agmon et al., 2016) . Therefore, the hydrated proton clusters in the acidified
$H_2O$+AS+DLT aerosol may preclude the substitution of $D_2O$ for $H_2O$ and give rise to a low diffusion extent. Such diffusion
limitation may provide a possible account for the long lifetime of certain ambient aerosols of which the unreacted core species
are protected from potential surface-sensitive phenomena such as cloud condensation nucleation (CCN) and ice nucleation
(IN) activities (Zhang et al., 2019; Adachi and Buseck, 2008; Kanji et al., 2019; Yu et al., 2019) .

## 5    Conclusions

In this work, we characterize the water diffusion process in single suspended phase-separated aerosols via a self-constructed
laser tweezers Raman spectroscopy system. The recorded Raman spectra of the aerosols is used to both detect their morphology
and observe the exchange of $D_2O$ and $H_2O$ molecules. The results of core-shell aerosols show that water molecules can pass
through the organic shell and diffuse into the particle bulk, where the diffusion rate depends on the types of the organic
compounds. Contrastingly, the partly engulfed and homogenous aerosols have higher diffusion rates. The results of the acidified
$H_2O$+AS+DLT aerosol show that surplus protons can improve water diffusion in the aerosol. It may be attributed to the surface-
active effect of the protons. Besides, the incomplete diffusion is observed in all the three types of aerosols with different
morphologies. By measuring the water diffusion coefficients and diffusion extents with a modified Fickian diffusion model,
we found that 65%~75% of the total $H_2O$ molecules in the phase-separated aerosols are substituted by $D_2O$ molecules, which
implies that certain molecule clusters form in the aerosol.

More works on the reactive uptake of gas molecules into the phase-separated aerosols should be done in the future. Besides,
the sizes of the droplets studied here are 4~10 $\mu$m and the techniques for detecting water diffusion in smaller phase-separated
droplets are imperative to be developed in the future.

*Code and data availability.* The datasets generated during this study are available at Peking University Open Research Data Platform:
https://doi.org/10.18170/DVN/LJMWYV.

*Author contributions.* Yu-Kai Tong proposed the idea of the project, performed the experiments, conducted the data analysis, and led in
writing the manuscript. Anpei Ye contributed to funding the research, constructed the optical tweezer system, provided the instruction on the
experiment and revised the manuscript. Zhijun Wu and Min Hu discussed the methodology and revised the manuscript.



*Competing interests.* There are no conflicts to declare.

*Acknowledgements.* This work was supported by the National Natural Science Foundation of China (U19A2007, 32150026 and 92043302).



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
