# Peer review of "Measurement Report: Water diffusion in single suspended phase-separated aerosols"

_EGUsphere, 2023_

## Author Comment (AC1)

**Dear Editors and Reviewers:**

Thanks for your letter and the reviewers' comments concerning our manuscript entitled "Measurement Report: Water diffusion in single suspended phase-separated aerosols" (Manuscript ID: egusphere-2023-1346). Those comments are all valuable and helpful for revising and improving our paper, as well as the important guiding significance to our researches. We have studied comments carefully, and revised the article by supplementing corresponding interpretations, conducting theoretical analysis, and adding more illustrations according to the reviewers' suggestions. Accordingly, the framework of this manuscript has been adjusted to systematically show the results that fits the theme. The point-by-point responses are as follows, and the corrections excerpted from the main article are highlighted by yellow. We hope the corrections can meet with your approval.

**Responds to the reviewers' comments:**

**Reviewer 1:**

**#1.** Can you show H2O/D2O exchange for just aqueous AS (or LiCl if efflorescence is an issue)? This would give some idea of the response time in your cell when switching between H2O and D2O. For instance, the characteristic time reported in Fig. 3 is probably just the time required for the cell to switch between H2O and D2O.

**Answer:**

The experiment for just aqueous AS may be not necessary. As shown in supplementary Fig. S1, the cell we used herein had a height of 3.5 cm, an inner diameter of 3 cm, and an outer diameter of 5 cm. Hence, the volume of the cell was less than 24.74 $cm^3$. The gas-washing bottle used as a bubbler in this work had a volume of 100 ml and contained 30 ml $D_2O$. The total sccm (standard cubic centimeter per minute) of dry and wet $N_2$ used herein was 100. Considering that RH = 60%, the maximum of the time required for the cell to switch between $H_2O$ and $D_2O$ was $(24.74 + 70)/(100 \times 60\%) = 1.58$ min. The observed characteristic time for aerosol spectra shown in Fig. 3 was 1.45 min. Thus, $D_2O$ had started to diffuse into droplet before gas exchange from $D_2O$ to $H_2O$ completed.

We have supplemented corresponding interpretations in the revised manuscript, as follow

"Volume of the aerosol chamber was < 24.7 $cm^3$ (Fig. S1). The gas-washing bottle used as a bubbler herein had a volume of 100 ml and contained 30 ml $D_2O$. The total flux of dry and wet $N_2$ used herein was 100 sccm. Thus, at RH = 60%, the maximum time required for the chamber vapor to switch between $H_2O$ and $D_2O$ was 1.58 min, which can be identified as the response time of the chamber (formally $\tau_{cell}$)." (Section 2.1)

"However, $\tau_{OD} < \tau_{cell}$ implies that $H_2O$ had started to diffuse into the droplet before gas exchange from $D_2O$ to $H_2O$ completed. Hence, water diffusion in homogenous CA droplet may be even faster than Fig. 3 showed." (Section 3.2)

**#2.** Throughout the paper you have this issue with \phi_{OD} not going to 0 or 1 as the experiment

time becomes very long. This leads to some fairly implausible proposals (in my opinion) towards the end of the discussion. Isn't it simply the well-known issue that strong WGMs heavily distort vibrational bands, making their use in quantitative work difficult? The classic example is Anal. Chem. 2005, 77, 7148-7155 where the authors, who had previously been using an EDB, switched to glass slides because of this very issue (they state this in the introduction to that paper).

Although I suppose the argument given on lines 205-207 that there is condensation of the species being replaced somewhere on the walls is also possible.

**Answer:**

As you mentioned, Wang et al. [Anal. Chem. 2005, 77, 7148] used Raman spectra to investigated the evolution of ion pairs during hygroscopic process of $MgSO_4$ droplets. Their focus was the intensity of certain sharp Raman peaks. However, in this work, we used Raman band area rather than peak intensity to trace the $H_2O$ or $D_2O$ concentration during water diffusion. As shown in Fig.2, the O-H band and O-D band were quite broad and the WGM peaks were pretty narrow. Thus, although WGMs distorted band silhouette, the area contribution of WGMs was trivial.

**#3.** Can you list the fitted droplet size where applicable, e.g. in Fig. 3.

**Answer:**

The particle radius of some studied droplets (e.g., $H_2O$+AS+DLT) has been presented in the manuscript (Fig. 6). The fitted radius of droplet shown in Fig.3 ($D_2O$+CA) is presented below.

[Figure]

**#4.** Why are the WGMs shifting in Fig. 3. Isn't the droplet in equilibrium with the surroundings? Is some other parameter changing?

**Answer:**

The experiments were conducted at room temperature (295$\pm$0.5 K). After capturing one aerosol droplet, the aerosol inlet was sealed immediately to maintain a constant RH in the chamber. No experimental parameters changed during the observation.

Nonetheless, because CA is a semi-volatile organic, the co-volatilization of CA and water may

slightly change droplet size, resulting in the shifting WGMs. Furthermore, spontaneous surface fluctuations, e.g. thermally induced capillary waves, existed in the optically trapped droplet (see [Endo et al., J. Phys. Chem. C 2018, 122, 20684; Pigot et al., Anal. Chem. 2012, 84, 2557; Chung et al., Anal. Chem. 2017, 89, 8092]), which may also contribute to the unstable WGMs. We have supplemented corresponding interpretations in the revised manuscript, as follow

"Since CA is a semi-volatile organic, the co-volatilization of CA and water may slightly change droplet size, resulting in the shifting WGMs. Furthermore, spontaneous surface fluctuations (e.g. thermally induced capillary waves) existing in the optically trapped droplet may also contribute to the unstable WGMs." (Section 3.2)

**#5.** What are the uncertainties associated with the measured \tau reported in Table 1? Were any of these measurements repeated more than once?

**Answer:**

The uncertainties of $\chi$ and $D_w$ were the parameter errors which were obtained by fitting temporal variation of $\phi_{OD}$ (see supplementary Fig. S5) using the modified Fickian diffusion model (Eq. 3). $D_w$ is a process parameter that characterizes water diffusion process, the determination of which is highly time-consuming. Thus, we did not conduct multiple measurements and average to retrieve the uncertainties. The static parameters, e.g., RH and droplet size, were retrieved from multiple measurements.

**#6.** Awkward writing:

"surplus protons are considered to have appreciable impacts..." This is a weird way of saying low pH.

"panoramically presents a panorama"

"equilibrated in D2O in ambiance"

**Answer:**

Thanks for these comments. We have changed these expressions in the revised manuscript.

If you have any query, please do not hesitate to contact me at the address below.
Thank you and best regards.

Yours sincerely,
Tong  Yu-Kai
E-mail:  tong.y.k@pku.edu.cn

Corresponding author:
Ye  Anpei
E-mail: yap@pku.edu.cn

---

## Author Comment (AC2)

**Dear Editors and Reviewers:**

Thanks for your letter and the reviewers' comments concerning our manuscript entitled "Measurement Report: Water diffusion in single suspended phase-separated aerosols" (Manuscript ID: egusphere-2023-1346). Those comments are all valuable and helpful for revising and improving our paper, as well as the important guiding significance to our researches. We have studied comments carefully, and revised the article by supplementing corresponding interpretations, conducting theoretical analysis, and adding more illustrations according to the reviewers' suggestions. Accordingly, the framework of this manuscript has been adjusted to systematically show the results that fits the theme. The point-by-point responses are as follows, and the corrections excerpted from the main article are highlighted by yellow. We hope the corrections can meet with your approval.

**Responds to the reviewers' comments:**

**Reviewer 2:**

**#1.** The figures showing the isotope exchange (Figure 3 in particular) indicate the WGM's move significantly during the process. Is this due to a change in the RH when switching between H2O and D2O? The SI indicates some correction to account for this, but it is unclear how effective this method is given the observations.

**Answer:**

The RH remained constant during the experiments. In Fig. 3, because CA is a semi-volatile organic, the co-volatilization of CA and water may slightly change droplet size, resulting in the shifting WGMs. Furthermore, spontaneous surface fluctuations, e.g. thermally induced capillary waves, existed in the optically trapped droplet (see [Endo et al., J. Phys. Chem. C 2018, 122, 20684; Pigot et al., Anal. Chem. 2012, 84, 2557; Chung et al., Anal. Chem. 2017, 89, 8092]), which may also contribute to the unstable WGMs. We have supplemented corresponding interpretations in the revised manuscript, as follow

"Since CA is a semi-volatile organic, the co-volatilization of CA and water may slightly change droplet size, resulting in the shifting WGMs. Furthermore, spontaneous surface fluctuations (e.g. thermally induced capillary waves) existing in the optically trapped droplet may also contribute to the unstable WGMs." (Section 3.2)

**#2.** The WGM's in the spectra are very broad, up to 1 nm in width. These features on particles of the size reported should be narrow, <<1 nm. Typically, in AOT systems, spectrometer resolution of <0.1nm are used to resolve 1st order modes. Can the authors confirm the grating and the resolution of their spectrometer?

**Answer:**

The spectrograph grating used herein was 1200 groove $mm^{-1}$ and the spectrometer resolution was ~4 $cm^{-1}$. To collect high SNR (signal-to-noise ratio) spectrum, we herein used a wide slit. Thus, the effective spectrometer resolution may be larger than 0.1 nm.

However, we used area of Raman band to trace $H_2O$ or $D_2O$ content within the droplet. The O-H band (27.3 nm in width) and O-D band (17.5 nm in width) were both disproportionately wider than the WGMs. Thus, the deviation caused by the spectrometer resolution may be trivial to the results of water diffusion examination.

**#3.** The spectrum for a core-shell particle will depend on the thickness of the shell, and whether the WGM's penetrate deeply enough to interfere with the internal interface. A core-shell particle retains spherical symmetry, so while the position of the WGM's in the spectrum may differ from a homogeneous particle, the peaks generally should appear well resolved. A deviation for spherical symmetry would lead to the spectrum degrading, which may occur if the shell thickness is not uniform.

**Answer:**

Thanks for this comment. As you said, the shell thickness has appreciable influences on aerosol spectrum. To be more rigorous, we have supplemented corresponding statements in the revised manuscript, as follow

"The shell thickness has appreciable influences on aerosol spectrum. The weak WGMs may be caused by bad sphericity of the droplet, such as non-uniform shell thickness. The droplet with good spherical symmetry and deep WGMs penetration can induce well-resolved WGM peaks." (Section 2.2)

**#4.** Were diffusion coefficients measured at lower RH for a well-characterized system, such as CA, to validate the accuracy against previous data?

**Answer:**

Thanks for this suggestion. We have conducted supplementary experiment to determine $D_w$ of CA under low RH conditions. The result of water diffusion in H2O+CA droplet at RH = 20% was shown below,

[Figure]

It showed that $\tau_{OD} = 13.67$ min at RH = 20% and was larger than 1.45 min at RH = 60% (see Fig. 3 in the main text), indicating that water diffusion was retarded under low RH conditions. The $D_w$ of $H_2O$+CA droplet (RH = 20%) measured in this work was close to the result of [Nadler et al., Phys. Chem. Chem. Phys. 2019, 12, 15062] (Table S1). It validates the performance of our system. We have added these interpretations in the revised supplementary information (Section S6).

**#5.** What is the chamber response time? How much does this vary due to the nebulization process depositing more material into the chamber? Was the chamber cleaned out between measurements?

**Answer:**

The response time of the aerosol chamber was 1.58 min which has been interpreted in the revised manuscript, as follow

"Volume of the aerosol chamber was < 24.7 cm³ (Fig. S1). The gas-washing bottle used as a bubbler herein had a volume of 100 ml and contained 30 ml $D_2O$. The total flux of dry and wet $N_2$ used herein was 100 sccm. Thus, at RH = 60%, the maximum time required for the chamber vapor to switch between $H_2O$ and $D_2O$ was 1.58 min, which can be identified as the response time of the chamber (formally $\tau_{cell}$)." (Section 2.1)

The RH in the chamber was first modulated to maintain at a specified value (60% herein). After nebulizing desired mother solution and capturing one aerosol droplet, the nebulization was stopped, and the aerosol inlet was sealed immediately to maintain a relatively isolated

environment in the chamber. The trapped droplet was allowed to rest in the chamber vapor for 10~15 min. In this period, the droplet can achieve an equilibrium with the surrounding gas and the surplus deposited material. Then, we launched the isotope exchange experiment. Thus, the deposited material had no influence on chamber response time. After one experiment finished, the chamber was disassembled and cleaned using an ultrasonic cleaner.

We have added corresponding interpretations in Section S1 in the revised supplementary information.

**#6.** How was the oleic acid mixture nebulized, given the insoluble nature of oleic acid? How much OA was in the levitated particles? Previously, OA has been introduced as a vapor into AOT's and allowed to condensed on existing particles (such as NaCl, in the work of Dennis-Smither et al. (https://doi.org/10.1029/2012JD018163)).

**Answer:**

The $H_2O$+AS+OA droplet was generated by nebulizing a mixed solution containing of AS, OA and water. The mixture was fully shaken to form an aqueous OA emulsion before nebulization. Thus, a number of OA-inclusions would be contained in the nascent droplet. In our experiment, the OA content in the droplet was random. If the droplet spectra cannot support a partly engulfed morphology, the droplet will be released until one spectra-confirmed partly engulfed droplet is captured. Then, water diffusion in such droplet was examined.

**#7.** Were measurements performed for water diffusion in 1,2,6-hexanetriol alone, under the same RH conditions? Based on its hygroscopicity and fluidity, it seems unlikely major diffusion limitations would be observed for water in this system. Does the presence of ammonium sulfate in particles in the present work lead to some kind of change in the rheology that leads to more diffusion limitations?

**Answer:**

Previously, we have already tried to examine water diffusion in droplet comprising 1,2,6-hexanetriol (HEX) only. However, the solubility of HEX was lower than that of inorganics (e.g., ammonium sulfate). Compared with HEX+AS droplet at the same RH level, the droplet comprising HEX only would shrink and result in a lower water content and a smaller equilibrium radius. The final radius of HEX droplet was lower than the minimum radius applicable to our LTRS system. In other words, HEX droplet cannot be trapped for a sufficiently long time for observing water diffusion.

Furthermore, previous work has reported that the structure of HEX+AS droplet was considerably complex at RH = 60%, where HEX-rich shell, AS-rich core, and more concentrated AS inclusions in the core coexisted [Ma et al., Atmos. Chem. Phys. 2021, 21, 9705]. The force between AS and water molecular, particularly in the case of concentrated AS inclusions, was highly stronger than that between HEX and water. Bound water may form under this circumstance. The substitution of $D_2O$ for $H_2O$ needed to overcome bound water force, which may be quite difficult because the solute concentration in AS inclusions was far beyond its solubility limit. This may also contribute to the slow water diffusion in HEX+AS droplet. Moreover, the water diffusion during humidifying or drying process was driven by RH changes; however, the diffusion investigated herein was translational

water diffusion at constant RH level, the weaker driven force of which may magnify the diffusion retardation extent.

We have supplemented corresponding interpretations in Section 3.4 in the revised manuscript.

**#8.** How much oleic acid is present in the particles? From the SI figure, the coating is small compared to the size of the droplet and the reduced surface area of the aqueous phase relative to a pure aqueous droplet would be small.

**Answer:**

Thanks for this comment. In the answer of Question **#6**, we interpreted that the OA content in the droplet was random. Volume ratio of the OA+AS mother solution was 4/1 (aqueous AS/OA). In Fig. 4, the quenched spectral WGMs implied an appreciable OA coating. To be more rigorous, we revised the schematic diagram in Fig. S2.

**#9.** Table 1 indicates the addition of acid leads to homogeneous morphology, but the rate of diffusion is still much slower than for a homogeneous system. Further the influence of lowered pH in this system would lead to protonated organic molecules that would be less polar, rather than more polar, suggesting miscibility should decrease rather than increase. Perhaps the pH influences the proton exchanges rates on the alcohol groups, which affects the timescales?

**Answer:**

Aerosol pH was reported to have significant impacts on liquid-liquid phase separation. However, the impacts were different and related to the organic component. Previous work found that the separation relative humidity (SRH, the RH level at which LLPS occurs) of some organic acids (e.g., 3-methylglutaric acid) increased as the aerosol pH decreased, while the SRH of other organics (e.g., polyols) decreased as the pH decreased [Losey et al., J. Phys. Chem. Lett. 2016, 7, 3861; Losey et al., J. Phys. Chem. A 2018, 122, 3819]. In the case of DLT+AS droplet, decreasing pH would inhibit the occurrence of phase separation. From a fundamental physical chemistry perspective, the fluctuations of local solute concentration will lead to thermodynamic instability in droplet structure and induce phase separation subsequently. In this work, the added sulfuric acid increased droplet viscosity, the plasticizing effect of which may limit concentration fluctuations and promote homogeneity. We have revised corresponding statement in Section 3.4 in the update manuscript. The slower water diffusion in the acidified droplet may be caused by the difficult substitution of $D_2O$ for $H_2O$ resulting from the already formation of hydrated protons ($H_3O^+$) or the so-called bound water. The characteristic diffusion time herein does not merely reflect the hygroscopicity capability of the droplet. What it directly reflects is diffusion rate of molecules in the size of water molecule into such droplet. Hygroscopic process may be prompt in neutral homogenous droplet. However, the equilibrium water vapor pressure at the surface of acidified homogenous droplet may be different from that of neutral homogenous droplet at the same RH level. Because lower vapor pressure means weaker driven force of diffusion, water molecule substitution in the droplet with low pH presented here was not that fast.

**#10.** For the core-shell fitting, can the authors provide simulated spectra to compare with the

measured core-shell spectra? More details on the use of this and the outputs would be useful.

**Answer:**

The core-shell model we used was developed by [Vennes et al., J. Opt. Soc. Am. A 2019, 36, 2089]. The inputs of this model were spectral WGM locations, initial guess for core/shell refractive index (RI), and range of radius ratios that are searched. The major outputs of the model were WGM locations, core radius, and particle radius. The parameters used herein were 1.33 for core RI initial guess, 1.6 for shell initial guess, and 0.5~0.99 for core/particle radius ratio. The comparison between simulated spectrum and recorded spectrum in Fig. 5 at time = ~93.5 min is presented below. It shows that the core-shell model fits the recorded spectrum very well. We have added corresponding interpretations in Section S7 in the revise supplementary information to show details on the use of this model.

[Figure]

If you have any query, please do not hesitate to contact me at the address below.
Thank you and best regards.

Yours sincerely,
Tong  Yu-Kai
E-mail:  tong.y.k@pku.edu.cn

Corresponding author:
Ye  Anpei
E-mail: yap@pku.edu.cn

---

## Author Response (AR2)

**Dear Editor:**

Thanks for your letter and the reviewers' comments concerning our manuscript entitled "Measurement Report: Water diffusion in single suspended phase-separated aerosols" (Manuscript ID: egusphere-2023-1346). Those comments are all valuable and helpful for revising and improving our paper, as well as the important guiding significance to our researches. We have studied comments carefully, and revised the article by supplementing corresponding interpretations, conducting theoretical analysis, and adding more illustrations according to the reviewers' suggestions. Accordingly, the framework of this manuscript has been adjusted to systematically show the results that fits the theme. The point-by-point responses are as follows, and the corrections excerpted from the main article are highlighted by yellow. We hope the corrections can meet with your approval.

**Responds to the editor's comments:**

**#1.** The revision does not yet convincingly address and incorporate the reviewers' questions and suggestions into the main text. Please adjust your responses to the following reviewer comments to be reflected in the revised manuscript: Reviewer 1's comments #2 and #3, Reviewer 2's comments #2, #4, #6, #8, and #9. In your point-by-point response, indicate clearly how you addressed the reviewer comments.

**Answer:**

We added new sentences and figures in the revised manuscript to response more directly to the reviewers' comments.

For Reviewer 1's comment #2,

"The O-H band and O-D band were quite broad while the WGM peaks were pretty narrow (Fig. 2); the fractional concentration of $D_2O$ was retrieved from Raman band area rather than peak intensity, thus the interference of WGMs to $\phi_{OD}$ was trivial." (Section 3.1, Line 146-148)

For Reviewer 1's comment #3, we have added the fitted droplet size in the update Fig. 3.

For Reviewer 2's comment #2,

"The spectrograph grating used herein was 1200 groove $mm^{-1}$ and the spectrometer resolution was ~4 $cm^{-1}$." (Section 2.1, Line 82) and Section 3.1 Line 146-148.

For Reviewer 2's comment #4,

"The water diffusion of CA droplet at lower RH (20%) was shown in Supplementary Fig. S6. The retrieved $\tau_{OD}$ was 13.67 min which was larger than that at RH = 60%, indicating that water diffusion was retarded under very low RH conditions. This result agreed well with previous works (Supplementary Tab. S1), validating the performance of our system." (Section 3.2, Line 171-174)

For Reviewer 2's comment #6,

"In this work, the $H_2O$+AS+OA droplet was generated by nebulizing a mixed solution containing of AS, OA and water. Volume ratio of the OA+AS mother solution was 4/1 (aqueous AS/OA). The mixture was fully shaken to form an aqueous OA emulsion before nebulization. Thus, a number of OA-inclusions would be contained in the nascent droplet. The OA content in the droplet herein was random. If the droplet spectra cannot support a partly engulfed morphology, the droplet will be released until one spectra-confirmed partly engulfed droplet is captured." (Section 3.3, Line 178-

182)

For Reviewer 2's comment #8,

"The volume ratio of aqueous phase and hydrophobic phase in the trapped droplet cannot be preset because of the stochastic mixing of OA emulsions and water during nebulizing." (Section 3.3, Line 190-191)

For Reviewer 2's comment #9,

"Aerosol pH was reported to have significant impacts on liquid-liquid phase separation. However, the impacts were different and related to the organic component. Previous work found that the separation relative humidity (SRH, the RH level at which LLPS occurs) of some organic acids (e.g., 3-methylglutaric acid) increased as the aerosol pH decreased, while the SRH of other organics (e.g., polyols) decreased as the pH decreased [Losey et al., J. Phys. Chem. Lett. 2016, 7, 3861; Losey et al., J. Phys. Chem. A 2018, 122, 3819]. From a fundamental physical chemistry perspective, the fluctuations of local solute concentration will lead to thermodynamic instability in droplet structure and induce phase separation subsequently. In this work, the added sulfuric acid increased droplet viscosity and reduced its liquidity, which may limit concentration fluctuations and promote homogeneity." (Section 3.4, Line 268-275)

"Moreover, the existence of bound water may limit water molecule evaporation and reduce the equilibrium water vapor pressure at the surface of acidified homogenous droplet (solute effect). While the surrounding moisture was switched from $H_2O$ to $D_2O$, $H_2O$ molecules within the droplet began to evaporate into the gas to maintain its vapor pressure equal the equilibrium pressure. Lower equilibrium vapor pressure means weaker driven force of diffusion, thus water molecule substitution in the droplet with low pH was not as fast as in neutral homogenous droplet." (Section 4, Line 316-321)

**#2.** Both reviewers commented on the shifting of whispering-gallery modes (WGMs). The response is rather brief and not sufficiently substantiated by experimental evidence. Please expand your response to this important point brought up by the reviewers.

**Answer:**

We supplemented more interpretations for this problem in the response letter to reviewers and in the revised manuscript in Line 155-163, as follow.

CA is a semi-volatile organic. Thus, after an CA aerosol was trapped, CA would volatilize from the droplet, inducing a decreasing solute concentration. However, in our experiments, the ambient RH maintained constant. To re-achieve an equilibrium with the surrounding moisture, the droplet would also evaporate water to maintain the solute concentration. Therefore, the co-volatilization of CA and water may slightly change droplet size. Such change of droplet radius was validated in the update Fig. 3 where the radius reduced by ~100 nm during 16 min. The WGMs were pretty sensitive to droplet size, thus the co-volatilization of CA and water and the induced changing size may result in the shifting WGMs. Furthermore, some spontaneous surface fluctuations, e.g. thermally induced capillary waves, existed in the optically trapped droplet (see [Endo et al., J. Phys. Chem. C 2018, 122, 20684; Pigot et al., Anal. Chem. 2012, 84, 2557; Chung et al., Anal. Chem. 2017, 89, 8092]). Surface fluctuations disturbed the standing wave at the interface between droplet and air, which may also contribute to the unstable spectral WGMs.

**#3.** Please thoroughly discuss the reasons and implications of \phi_{OD} not going to 0 or 1 (Reviewer 1, comment #2).

**Answer:**

We supplemented more interpretations for this issue in the response letter to reviewers, as follow. $\phi_{OD}$ did not go to 0 or 1 meant that $D_2O$ in the trapped droplet cannot be completely substituted by $H_2O$. It can be explained by two possible reasons. (i) The aerosol optical tweezer system has an inevitable experiment limitation. In the beginning of one experiment, an aerosol train was consecutively introduced into the chamber until one droplet was captured from the train. The other aerosols may condense on the walls of the chamber and tubes. Therefore, although the vapor was switched from $H_2O$ to $D_2O$, the $H_2O$ molecules from the condensate aerosols may interfere water displacement extent, making $\phi_{OD}$ unable to go to 0 or 1. (ii) Some water molecules within the droplet exist in the form of bound water, e.g. combined water in supersaturated AS inclusions and hydrated protons in acidified aerosols. Water displacement needed to overcome the bound water force which was much stronger than the diffusion force, making complete molecules substitution impossible.

**#4.** The discussions in the manuscript could use better referencing. It is sometimes not clearly indicated what is a known fact (provide reference to literature), an observation (provide reference to data), or speculation (indicate in your wording).

**Answer:**

Thanks for this problem. We have used more rigorous descriptions to distinguish these three circumstances (see Line 98, 126, 156, 160, 188, etc.).

**#5.** Consider adding the H2O + CA data (new in supplement) to main manuscript.

**Answer:**

This manuscript focused on water diffusion in phase-separated aerosol droplets. Since $H_2O$+CA is not a phase-separated droplet, we think it is appropriate to put this discussion in the supplementary information. We have added enough corresponding interpretations on $H_2O$+CA data in the manuscript (Line 171-174).

**#6.** Response to reviewer 2, comment #3: Please discuss in more detail how this may affect the conclusions of the manuscript.

**Answer:**

This comment provided an inspiring interpretation for the weak WGMs in spectrum of core-shell droplet (see Fig. S2 B). It supported the identification of core-shell droplet that spectra containing weak but noticeable WGMs indicated a core-shell morphology. It also provided a possible reason for the unstable WGMs in spectra of core-shell droplet. Thus, this comment supported the conclusion of this manuscript and made the discussions more comprehensive. We added this comment in Section 2.2 (Line 105-108) to make the statement more rigorous.

**#7.** l. 249 of revised manuscript: "the added sulfuric acid increased droplet viscosity, the plasticizing effect of which may limit local concentration fluctuations and promote homogeneity" - Please provide proper reference or indicate speculation. Also clarify which molecule's plasticizing effect is referred to here.

**Answer:**

This statement is flawed. We misunderstood the plasticizing effect of water. We revised this statement as follow,

"The added sulfuric acid increased droplet viscosity and reduced its liquidity, which may limit concentration fluctuations and promote homogeneity." (Section 3.4)

**#8.** l. 250 of revised manuscript: "Notably, what the characteristic diffusion time directly mirrors is diffusion rate of molecules in the size of water molecule into such droplet." - The meaning of the sentence is unclear, please revise.

**Answer:**

This statement is inappropriate and redundant. We deleted it in the update revised manuscript.

**#9.** l. 252 of revised manuscript: "However, the equilibrium water vapor pressure at the surface of acidified homogenous droplet may be different from that of neutral homogenous droplet at the same RH level." - This is unclear. Please provide reference or more detailed explanation for this statement. Are you suggesting that the relationship of water activity and vapor pressure may differ?

**Answer:**

We did not suggest that the relationship of water activity and vapor pressure may differ. According to Raoult's law, the equilibrium water vapor pressure at the surface of a droplet (denoted by $p^0$) is influenced by the solute. Compared with a neutral droplet, a portion of water molecules in an acidified droplet were combined with protons in the form of hydrogen bond, preventing them from evaporating to the gas phase. Thus, the water vapor pressure at surface of an acidified droplet (denoted by $p_a^0$) was lower than that of a neutral droplet (denoted by $p_n^0$).

At an equilibrium state, $H_2O$ vapor pressure at the surface of a droplet (denoted by $p_d$) equals $p^0$. While the surrounding moisture was switched from $H_2O$ to $D_2O$, $p_d$ decreased to zero. $H_2O$ molecules within the droplet began to diffuse into the gas phase to maintain $p_d = p^0$. However, since $p_a^0 < p_n^0$, the diffusion driven force in an acidified droplet was weaker than that in a neutral droplet, yielding a lower water molecules substitution rate.

We added these details in the update revised manuscript in Line 316-321.

**#10.** It is crucial that you consult the reviewer comments in detail and make more comprehensive adjustments to your manuscript. Please understand that failing to make these major revisions will likely result in the rejection of your manuscript. The revised manuscript will undergo another round of peer review to assess whether the issues have been adequately resolved.

**Answer:**

Thanks for this comment. The reviewers' opinions were pretty instructive for our work. We considered these comments carefully and revised the manuscript in detail. We hope the corrections can meet with your approval.

If you have any query, please do not hesitate to contact me at the address below.
Thank you and best regards.

Yours sincerely,
Tong  Yu-Kai
E-mail:  tong.y.k@pku.edu.cn

Corresponding author:
Ye  Anpei
E-mail: yap@pku.edu.cn

---

## Author Response (AR3)

**Dear Editors and Reviewers:**

Thanks for your letter and the reviewers' comments concerning our manuscript entitled "Measurement Report: Water diffusion in single suspended phase-separated aerosols" (Manuscript ID: egusphere-2023-1346). Those comments are all valuable and helpful for revising and improving our paper, as well as the important guiding significance to our researches. We have studied comments carefully, and revised the article by supplementing corresponding interpretations according to the reviewers' suggestions. Accordingly, the framework of this manuscript has been adjusted to systematically show the results that fits the theme. The point-by-point responses are as follows, and the corrections excerpted from the main article are highlighted by yellow. We hope the corrections can meet with your approval.

**Responds to the reviewers' comments:**

**Reviewer 1:**

**#1.** I do not agree that the changes in the spectra shown in Figure 3 can be rationalized by evaporation of citric acid. It is much more likely to arise from a change in RH occurring during the H2O/D2O shift, evidenced by the rapid change in peak positions initially (in the first 2 mins), followed by a deceleration of the change over the next few minutes.

**Answer:**

You are right, the WGM shift may be induced by the RH change experienced by the droplet. While the moisture was switched from $D_2O$ to $H_2O$, the $H_2O$ needed ~1.6 min (chamber response time) to fill the chamber. However, during this period, dry nitrogen entered the chamber sustainedly while $H_2O$ molecules delayed in the bubbler bottle and gas tubes. The true RH experienced by the droplet may hence decrease, inducing the WGM shift.

We have supplemented these interpretations in the revised manuscript (Section 3.2, Line 160-164).

**#2.** Regarding diffusion rates in hex-AS-water system - I remain surprised by the observation that any diffusion limitations are observed. However, given some of the observations reported by Richards et al. (DOI: 10.1126/sciadv.abb5643), this may not be totally unexpected if some of the ammonium ions remain in the organic-rich phase and influence the viscosity through ion-molecule interactions.

**Answer:**

Thanks for providing this enlightening reference. According to Richards et al., the supramolecular ion-organic interactions may exist when aerosols contain organics (specifically those containing vicinal hydroxyl groups) and inorganic divalent ions, which produces internal cross-linking molecular networks. Such ion-organic networks may form in the shell of $H_2O$+AS+HEX aerosol and thus inhibited water diffusion.

We have supplemented these interpretations in the revised manuscript (Section 3.4, Line 255-258).

If you have any query, please do not hesitate to contact me at the address below.
Thank you and best regards.

Yours sincerely,
Tong  Yu-Kai
E-mail:  tong.y.k@pku.edu.cn

Corresponding author:
Ye  Anpei
E-mail: yap@pku.edu.cn